# Drivers of Spatial and Temporal Dynamics in Water Turbidity of China Yangtze River Basin

**Jian Li and Chunlin Xia ***

School of Remote Sensing and Mapping Engineering, Nanjing University of Information Science and Technology, Nanjing 210044, China
* Correspondence: chunlinxia163@163.com; Tel.: +86-13856559092

**Abstract:** The sustainable development of the water environment in the Yangtze River basin has become a critical issue in China. Turbidity is a comprehensive element for water quality monitoring. In this study, the baseline of water turbidity in the Yangtze River was constructed using 36 years of Landsat images from 1986 to 2021. The spatial and temporal dynamics of turbidity and its driving factors were explored. The results show that (i) the proposed Landsat-based turbidity model performs well, with a correlation coefficient ($R^2$) of 0.68 and a Root Mean Square Error (RMSE) of 7.83 NTU for the whole basin. (ii) The turbidity level in the Yangtze River basin is spatially high in the upper reaches (41.7 NTU), low in the middle reaches (30.9 NTU), and higher in the lower reaches (37.6 NTU). The river turbidity level (60.1 NTU) is higher than the turbidity in lakes and reservoirs (29.6 NTU). The turbidity in the Yangtze River basin shows a decreasing trend from 1986 to 2021, with the most significant decrease in the mainstream of the Yangtze River. Seasonally, the mean turbidity in the Yangtze River basin shows a "low in summer and high in winter" trend, but opposite trends were revealed for the first time in rivers and lakes, such as Dongting Lake, Poyang Lake, and Taihu Lake, etc. (iii) Natural factors, including precipitation and natural vegetation cover (woodlands, grasslands, and shrubs) could explain 58% of the turbidity variations, while human activities including impervious surfaces, cropland, and barren land are lower impact. Annual precipitation was negatively correlated with water turbidity, while cropland and barren land showed a significant positive correlation. The study is of great practical value for the sustainable development of the water environment in the Yangtze River basin and provides a reference for remote sensing monitoring of the water environment in inland water bodies.

**Keywords:** water turbidity; spatial–temporal dynamics; random forest; driving factors; Yangtze River Basin

## 1. Introduction

The Yangtze River is the longest river in China, an essential support for developing the Yangtze River Economic Belt, the integrated development of the Yangtze River Delta, and other national strategies [1]. It is an important line of defense for China's water resources' security and the basis for the economic and social development of the Yangtze River Basin [2]. It is important for China's economic and social development [3]. Since the 1980s, with rapid economic growth, industrialization, and urbanization, the water environment quality in the Yangtze River Basin has deteriorated [4]. In the 21st century, the state, government, and relevant departments have taken remediation measures. The water quality of the Yangtze River and Yangtze River Basin has improved significantly compared to before the remediation. However, due to the weak foundation and many historical debts, water ecology is still the main problem in the Yangtze River Basin, mainly manifested by serious soil erosion in the upper reaches, severe agricultural surface pollution, and eutrophication of water bodies in the middle and lower reaches, and shrinkage of some wetlands and lakes [5].

The conventional water quality monitoring method lays many sampling sections in the whole water area through manual sampling to obtain real-time field pollutant concentrations. The technique requires a lot of human and material resources, and some areas are challenging to sample, restricted by hydrological, climatic, and other natural conditions. Moreover, data obtained from the field sampling work are not a good reflection of the water quality information of the whole region. In recent years, the rapid development of remote sensing technology has been increasingly used for the dynamic monitoring of large-scale water quality information [6]. Remote sensing satellites have all-weather, all-round, large-scale real-time imaging and are not subject to natural geographical conditions. The application of remote sensing in water quality monitoring can effectively compensate for the shortcomings of traditional methods in temporal and spatial continuity [7]. In recent years, many scholars have analyzed the water environment of the Yangtze River Basin through remote sensing monitoring of water quality, and have conducted much research on the water quality of the mainstream of the Yangtze River and the lakes in the middle and lower reaches of the river. Standard remote-sensing water-quality monitoring inversion models include empirical, semi-empirical models [8], and machine learning models [9]. Studies have used various satellite data for different water quality indicators in other water bodies, including chlorophyll a concentration, suspended solids concentration, total phosphorus, total nitrogen, and turbidity. For instance, ocean color satellite data, including Terra/Aqua MODIS, Sentinel-2 MERIS, and Sentinel-3A OLCI, have been widely adopted for water quality monitoring using varied models such as exponential function model, neural networks, and random forest model [10–14]. Moreover, some land-oriented satellite sensors include Landsat TM/ETM+/OLI data, GF-1 WFV, HJ-1A/B, and HMS-2 for inland waters such as lakes, reservoirs, and rivers [15–22]. Despite the advances in remote-sensing monitoring of water quality in the Yangtze River Basin, the research on monitoring water quality at large spatial and temporal scales still needs to be completed.

The relationship between land use and water quality has been extensively studied and shown to impact water quality directly [23,24]. The analysis of this relationship is crucial for protecting watershed soil and water resources, and various methods, such as correlation analysis [25] and redundancy analysis [26], have been employed. For instance, Zhang et al. [27] conducted a study in the Three Gorges reservoir area using redundancy analysis and multivariate statistical analysis to examine the impact of land use on water quality. It should be noted that the results of different watersheds vary due to unique natural factors such as land use structure, topography, climate, and the soil's physical and chemical properties. Additionally, the impact of land use on water quality can be significantly different depending on the spatial scale of the study [28]. Zhang et al. [26] found that the watershed scale significantly impacted water quality in the Daning River Basin. Similarly, Xu Qiyu et al. [29] discovered that the riparian zone showed the most significant impact on water quality in the Ganjiang River in Yuanhe and Poyang Lake Basins, with forest land, paddy fields, and residential construction land having the most significant influence. Wang Yishu et al. [30] found that the riparian zone within 800 m of the Xijiang River significantly impacted on water quality. Wang J et al. [31] studied the relationship between land use and landscape pattern of the Danjiang River and water quality. They concluded that cropland and construction land have a negative impact on water quality. This was also confirmed by Yang Qiangqiang et al. [32] in their study of the Qingge River Basin, which showed that cropland and building land have a negative impact on water quality. In conclusion, studies of the relationship between land use and water quality at different spatial scales is crucial in optimizing land use and controlling nonpoint source pollution.

Turbidity is an essential parameter for monitoring river water quality [33], and it has a close relationship with suspended solids [34]. The analysis of turbidity allows us to understand the distribution of suspended matter or sediment in a water body. The changes in turbidity can provide an insight into the behavior of pollutants such as sedimentation, decomposition, and dispersion [21]. Hence, it is crucial to monitor the spatial distribution of turbidity to gain a comprehensive understanding of water quality. This study focuses on water turbidity variations in the Yangtze River Basin, using in situ observation data and Landsat series remote-sensing satellite images from 1986 to 2021. The study also examines drivers of the spatial and temporal variations of water quality parameters at the basin, sub-basin, and grid scales, based on the long-term remote sensing results. The results aim to provide a baseline for water environment management in the Yangtze River Basin and provide a reference for remote sensing monitoring of the water environment in inland water bodies. The step diagram of this paper is shown in Figure 1.

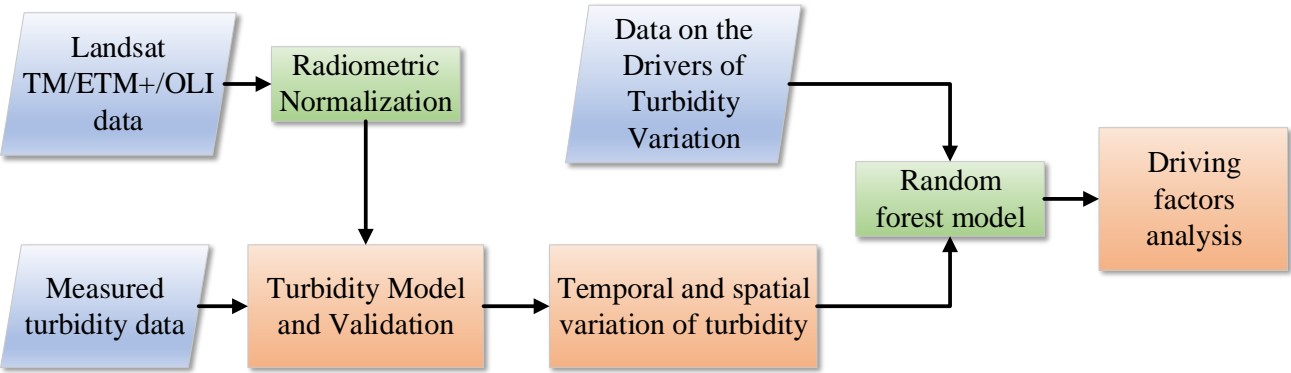

**Figure 1.** Step block diagram.

## 2. Study Area and Data

### 2.1. Overview of the Study Area

The Yangtze River Basin is situated between 24°30′ N to 35°45′ N and 90°33′ E to 122°25′ E, covering an area of 1.8 million km², accounting for approximately 20 percent of the total area of China [35]. The basin encompasses 11 provinces, cities, and autonomous regions, and is divided into 9 secondary and 45 tertiary sub-basins. The secondary sub-basins include the Yalong River Basin, Minjiang River Basin, Jialing River Basin, Yangtze River Main Current Basin, Hanjiang River Basin, Wujiang River Basin, Dongting Lake Basin, Poyang Lake Basin, and Taihu Lake Basin. The Yangtze River is divided into three reaches based on its geographical environment and hydrological characteristics. From Heyuan to Yichang in Hubei Province, the upper reach passes through high plateaus, mountains, and canyons with large drops, such as the Tongtian, Jinsha, and Three Gorges area, which exhibit characteristics of prominent highland mountain canyon rivers. The middle reach, from Yichang to Hukou, is winding and twisting with a relatively open water surface and a slower flow velocity, with numerous tributaries and lakes, including the Han River to the north and rivers such as the Xiangjiang, Zishui, Yuanjiang, Dongting Lake Basin, and Ganjiang, Fujiang, Xinjiang, Xiuzhushui in the Poyang Lake Basin. The lower reach, from Hukou to the mouth of the sea, is more than 800 km long, with an open water surface, slower velocity, and numerous tributaries, although not as large as those in the middle and upper reaches. The division of the Yangtze River Basin into multiple scales is important for a detailed study of hydrological changes in different basin areas and for analyzing the causes. The secondary and tertiary divisions of the basin are presented in Figure 2 and Table 1.

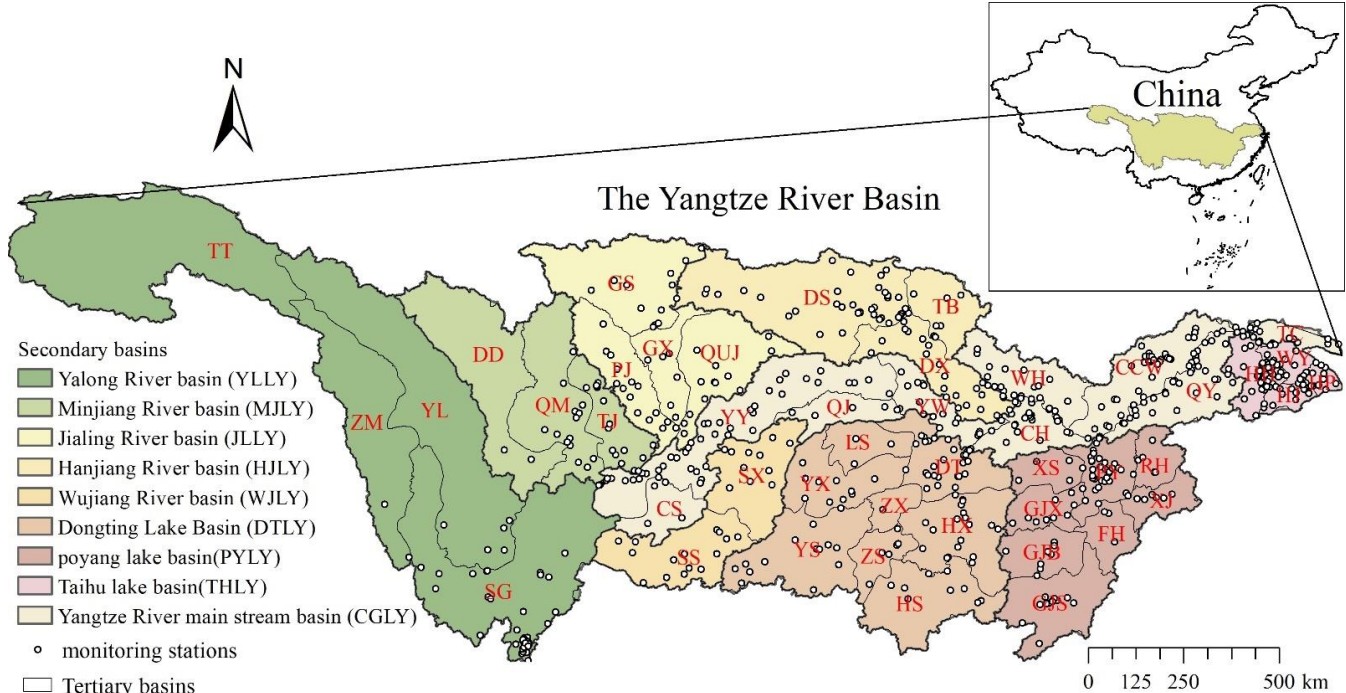

**Figure 2.** Map of the study area, including the distribution of secondary, tertiary, and water quality monitoring stations.

**Table 1.** Names, affiliations, number of monitoring sites, and measured turbidity of 3 rivers reach 9 second-level watersheds and 45 third-level.

| Reaches | Name of the Secondary Basin | Abbreviations of Tertiary Basin | Number of Sites |
|---------|----------------------------|--------------------------------|-----------------|
| Upper reaches of the Yangtze River | Yalong River Basin (YLLY) | TT | 0 |
| | | ZM | 2 |
| | | YL | 5 |
| | | SG | 23 |
| | Minjiang River Basin (MJLY) | DD | 2 |
| | | QM | 13 |
| | | TJ | 15 |
| | | PJ | 15 |
| | Jialing River Basin (JLLY) | GX | 6 |
| | | GS | 12 |
| | | QUJ | 13 |
| | Wujiang River Basin (WJLY) | SS | 15 |
| | | SX | 10 |
| | Yangtze River main stream Basin (CGLY) | CS | 5 |
| | | YY | 44 |

**Table 1.** *Cont.*

| Reaches | Name of the Secondary Basin | Abbreviations of Tertiary Basin | Number of Sites |
|---|---|---|---|
| Middle reaches of the Yangtze River | Hanjiang River Basin (HJLY) | DS | 36 |
| | | DX | 19 |
| | | TB | 7 |
| | | LS | 3 |
| | | YS | 14 |
| | | YX | 12 |
| | Dongting Lake Basin (DTLY) | ZS | 6 |
| | | ZX | 1 |
| | | DT | 22 |
| | | HS | 6 |
| | | HX | 19 |
| | | XS | 5 |
| | | GJX | 12 |
| | | GJB | 8 |
| | Poyang lake Basin(PYLY) | GJS | 11 |
| | | PY | 28 |
| | | FH | 3 |
| | | RH | 4 |
| | | XJ | 6 |
| | | QJ | 5 |
| | Yangtze River main stream Basin (CGLY) | YW | 18 |
| | | WH | 24 |
| | | CH | 21 |
| Lower reaches of the Yangtze River | Taihu lake Basin (THLY) | HH | 42 |
| | | WY | 12 |
| | | HJ | 14 |
| | | HP | 17 |
| | Yangtze River main stream Basin (CGLY) | CCW | 31 |
| | | QY | 34 |
| | | TC | 19 |

*2.2. Data*

2.2.1. Remote Sensing and In Situ Data for Turbidity Inversion

In this study, the remote sensing data sources used were Landsat 5 TM, Landsat 7 ETM, and Landsat 8 OLI surface reflectance data from 1986 to 2021. These data were obtained from USGS and processed using the Google Earth Engine (GEE) platform for radiometric calibration and atmospheric correction. The resulting data have a spatial resolution of 30 m. To ensure that the number of remote sensing images would not affect the results, the number of effective Landsat observations in the Yangtze River Basin were counted from 1986 to 2021, as shown in Figure 3. The results showed that the distribution of effective observations was uneven, with a greater concentration in the upstream and downstream areas and less in the midstream areas. The Wujiang River Basin had the least number of effective observations, with an average of 8 images per year, while the Yalong River Basin had the most, with 18 images per year. This indicates that there were enough Landsat observations between 1986 and 2021 to support the investigation of interannual variability in turbidity in the Yangtze River Basin.

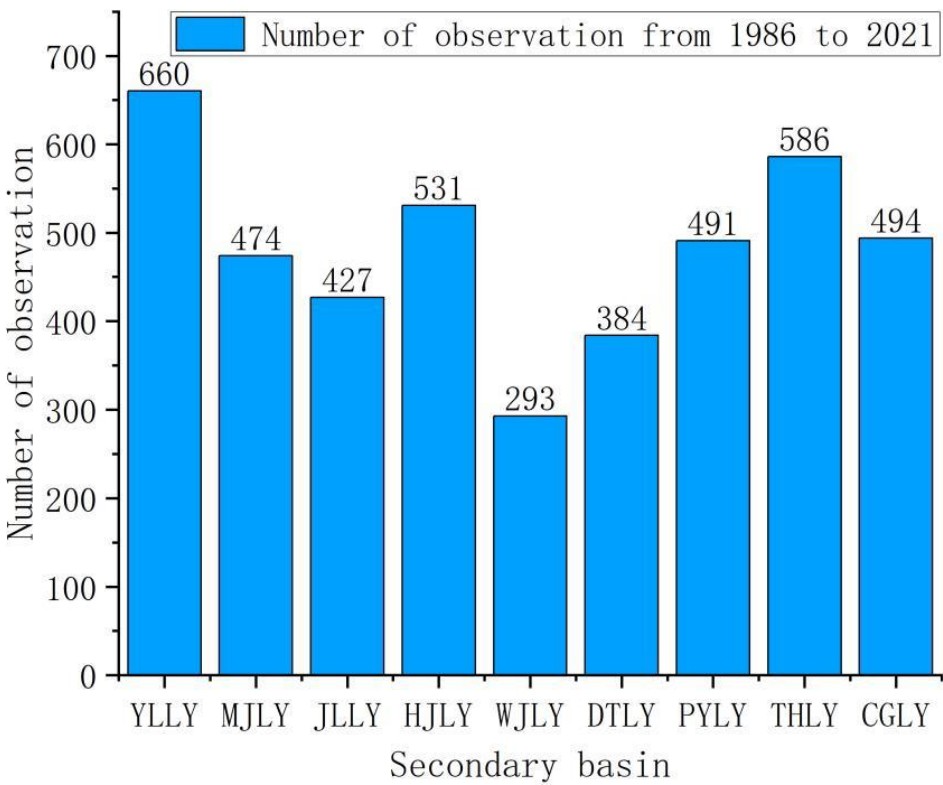

**Figure 3.** Effective Landsat observations in the secondary basin from 1986 to 2021.

In this study, the in situ water quality data were obtained from the China General Environmental Monitoring Station and the national surface water-quality monitoring network of the Yangtze River Basin cross-section monitoring results. The turbidity data from 639 water quality monitoring stations in the Yangtze River Basin in 2021 were analyzed to build and validate the turbidity inversion model. Figure 1 shows the distribution of these monitoring stations, which are almost widespread throughout the basin. It can be observed that the distribution of stations in large lakes and rivers is more concentrated.

2.2.2. Data on the Drivers of Turbidity Variation

The scouring effect of rainfall runoff has been found to impact on watershed water quality. The type of land use reflects the primary sources of nutrients and the soil erosion status of the watershed, which then affects the river water environment and, ultimately, the turbidity of the watershed. With increasing population and economic development, human activities, such as domestic and industrial pollution, pose a growing threat to water quality. Therefore, this study evaluates the contribution of various drivers to turbidity in the Yangtze River Basin, including both natural factors and human activities. Natural factors include precipitation and natural vegetation cover, such as woodlands, grasslands, and shrubs, in the watershed. Human activities include impervious surfaces, croplands, and barren land. The cropland cover reflects agricultural pollution, such as using fertilizers and pesticides. The barren land cover representing destruction of undeveloped areas due to human activities, such as mining, indicates soil erosion in the basin. Impervious surface cover, to some extent, reflects the impact of local urban pollution, such as domestic and industrial wastewater, on water quality.

Precipitation data from 1986 to 2021 were obtained from the Climate Hazards Group Infrared Precipitation Station Dataset (CHIRPS) through the Google Earth Engine (GEE) platform. As shown in Figure 4a, the annual precipitation in the upper river source area is the lowest, with less than 400 mm, and the average annual precipitation in the upper reaches is less than 800 mm. The precipitation in the middle and lower reaches is more significant, with an average annual precipitation of over 1300 mm, among which the

Poyang Lake Basin has the most abundant precipitation, exceeding 2000 mm annually. From a seasonal perspective, the precipitation in the dry period (from May to October) of each sub-basin is lower than that in the abundant period (from January to April, November, and December) [36]. The months of maximum precipitation are mainly concentrated from June to August.

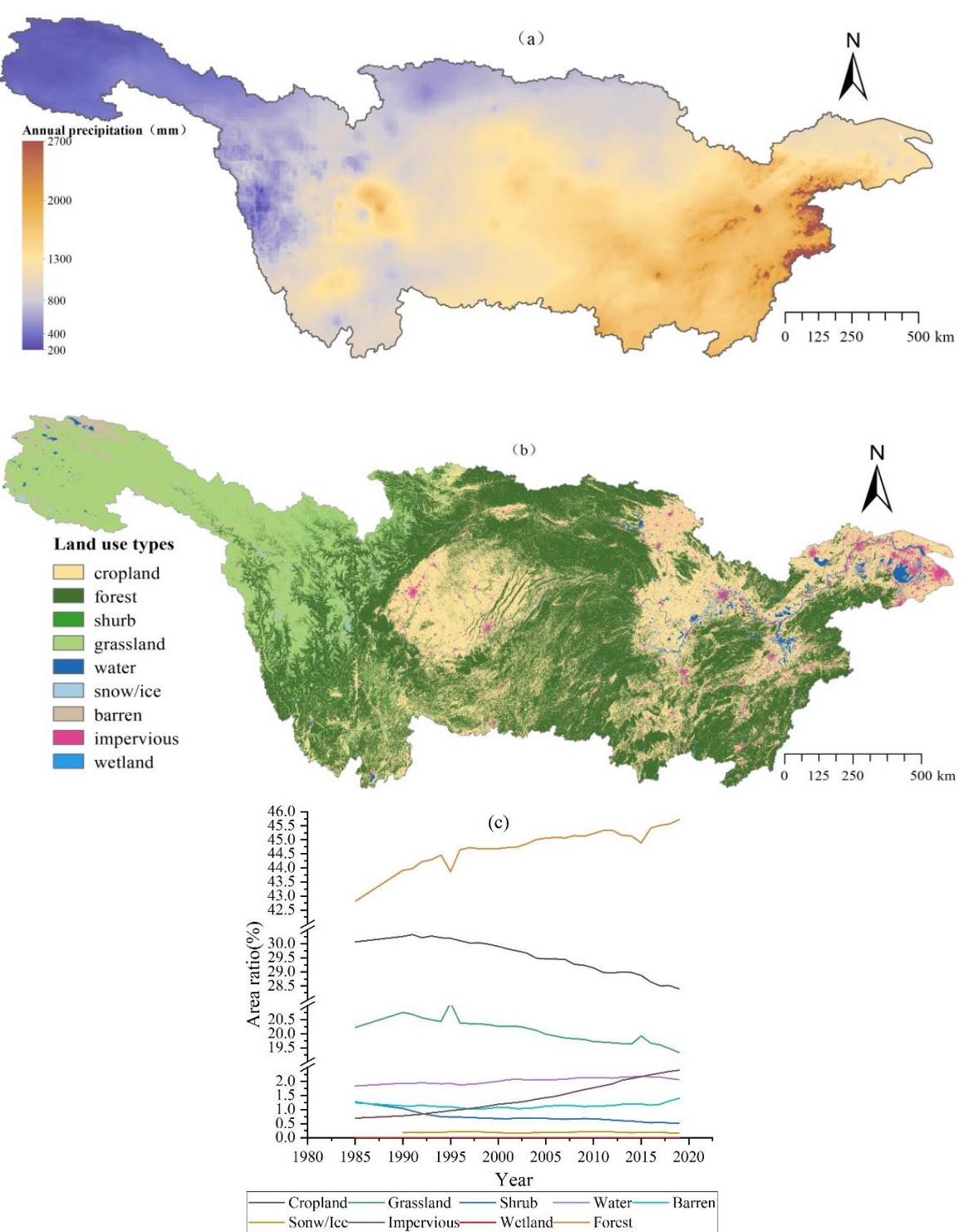

**Figure 4.** (**a**) The average annual precipitation of the Yangtze River Basin from 1986 to 2021; (**b**) Distribution of land use types in the Yangtze River Basin in 2019; (**c**) Area Changes of land use types in the Yangtze River Basin from 1985 to 2020.

Land cover data were obtained from the China Land Cover Dataset developed by the Wuhan University Institute of Remote Sensing. The dataset contains annual land cover information for China from 1985 to 2020 with a spatial resolution of 30 m and an accuracy rate of 80%. The data are classified into nine categories: cropland, forest land, shrubs, grassland, water bodies, snow/ice, wasteland, impervious surface, and wetland. The map of land use types (Figure 4b) shows that the primary land use types in the Yangtze River Basin are cropland, forest land, grassland, impervious surface, and water bodies. As shown in Figure 4c, a decrease in cropland and grassland areas from 1985 to 2020, accounting for 28.40% and 19.33% of the basin area, respectively. On the other hand, the forested land and impervious surfaces increased, covering 45.72% and 2.40% of the watershed area in 2020, respectively.

## 3. Methodology

### 3.1. Radiometric Normalization of Series Landsat TM/ETM+/OLI Data

In applying long-term remote sensing images for change detection or quantitative remote sensing, pseudo changes can sometimes occur in a single feature on the ground surface due to differences in radiation characteristics among various sensors, such as band range, central wavelength, and band response. To address this issue, this study employs the relative radiation normalization method proposed by Xiaomin Yu [37] to minimize these pseudo variations. High-quality Landsat 8 OLI images of the Yangtze River Basin were selected as the reference image, while quasi-synchronous Landsat TM/ETM+ images were normalized. The corresponding visible and near-infrared wavebands from the reference image and the normalized image were overlaid to calculate the coefficient of variation, with lower values indicating lower variations in the region, which were used as pseudo-invariant feature points. Using the pseudo-invariant feature point vector, the reflectance of each image band was extracted, and regression models for each band were obtained (as seen in Figure 5). The process is outlined in more detail below.

$$Y_t = a_t x_t + b_t \qquad (1)$$

where $x_t$ is the reflectance of the image to be normalized at band t, $y_t$ is the reflectance of the reference image at band t, and $a_t$ and $b_t$ are the slope and intercept of the regression equation at band t, respectively.

### 3.2. Turbidity Model and Validation

This study utilizes Landsat surface reflectance images and simultaneous in situ turbidity data to derive different turbidity inversion algorithms. The in situ measured turbidity values from 232 stations in the Yangtze River Basin in 2021 and the corresponding band reflectance combinations were used to construct these algorithms (as shown in Table 2), and the optimal band or band combination was identified [38]. The validity of the selected algorithm was verified using in situ measured data. The study finally adopted a linear model proposed by Yang Zhen et al. [39], using the combination of bands (green) and (red) as the independent variable [(green) − (red)]/[(red) + (green)]. The model yielded an $R^2$ of 0.68. The measured water turbidity data validation showed that the model had good inversion results, with a Pearson correlation coefficient of 0.76 and a root mean square error of 7.83 NTU (as seen in Figure 6).

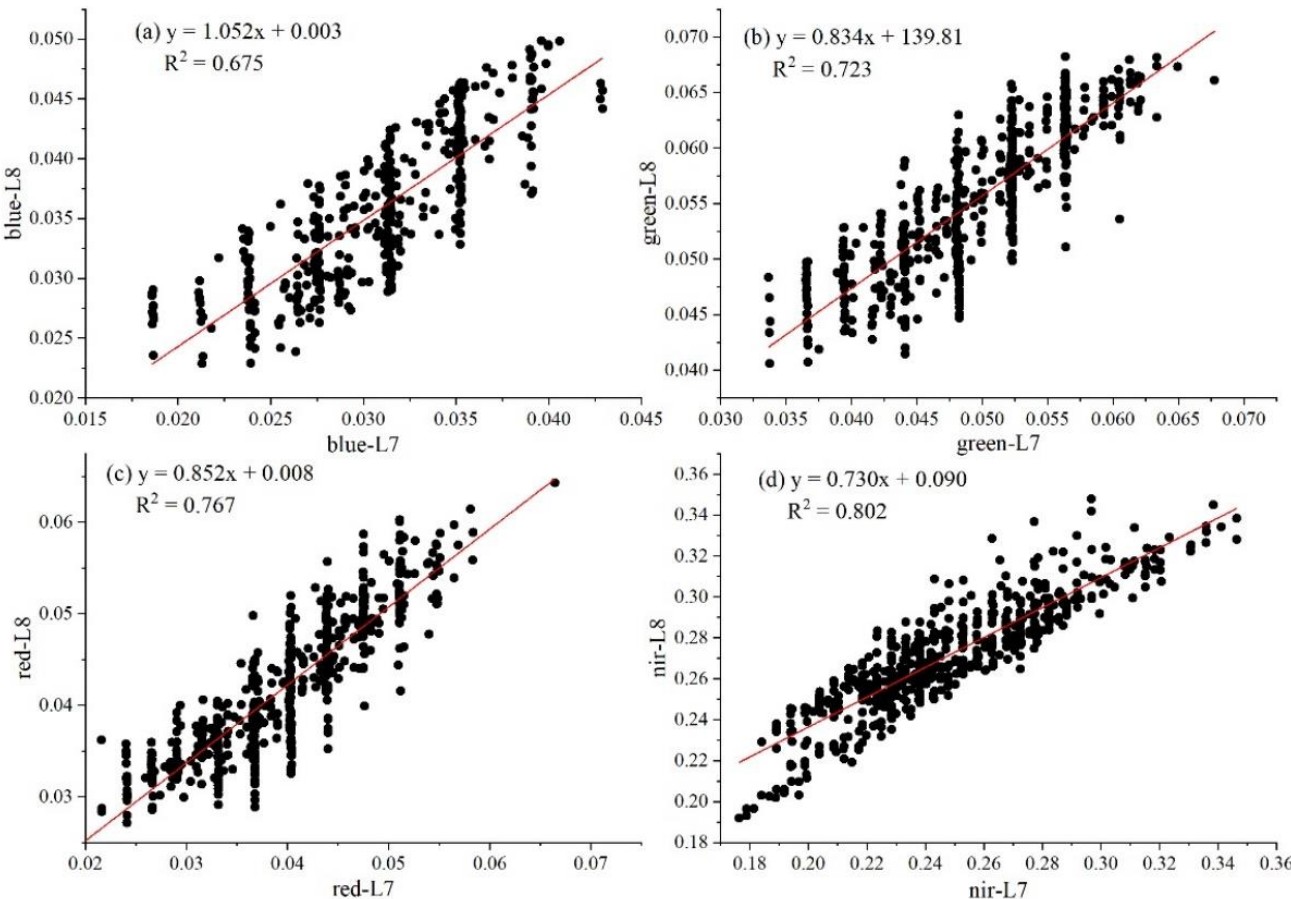

**Figure 5.** Regression analysis of pseudo-invariant feature points. (**a**) blue band, (**b**) green band, (**c**) red band, (**d**) nir band.

**Table 2.** Inversion algorithm of different turbidity.

|   | Remote Sensing Factor | Equations | R2 | Reference Model |
|---|---|---|---|---|
| 1 | $x = b(red)/b(blue)$ | $y = 0.91e^{2.82x}$ | 0.58 | Kratzer S. et al. [40] |
| 2 | $x = b(red)/b(green)$ | $y = 286.5x^2 - 333.8x + 103.9$ | 0.62 | Hou X. et al. [41] |
| 3 | $x = b(nir)$ | $y = 4725.5x^2 - 141.3x + 6.90$ | 0.46 | Petus C. et al. [42] |
| 4 | $x = (b(green) - b(red))/b((red) + b(green))$ | $y = 52.1e^{-9.64x}$ | 0.68 | Yang Z. et al. [39] |
| 5 | $b(blue)$, $b(green)$, $b(red)$, $b(nir)$ | Multiple linear regression | 0.50 | Maeda E.E. et al. [43] |

### 3.3. Driving Factors Analysis

We employed a random forest model to examine the drivers' response to turbidity levels. This model is capable of identifying the key variables affecting the response among a large number of covariate variables. The analysis was performed using the scikit-learn library in Python. We started by fitting the explanatory variables and the mean turbidity level in the Yangtze River Basin through a random forest regression approach. Then, we computed the feature importance to gauge the drivers' response. The feature importance reflects the contribution of each explanatory variable and is calculated through permutations of the variables. For each explanatory variable, the significance of its ranking is determined by the increase in the prediction's mean squared error (MSE) when that feature is ranked [44]. The rise in MSE for each explanatory variable is normalized and expressed as a percentage between 0 and 100%. A significant increase in MSE indicates a higher impact and contribution of the explanatory variable to the response.

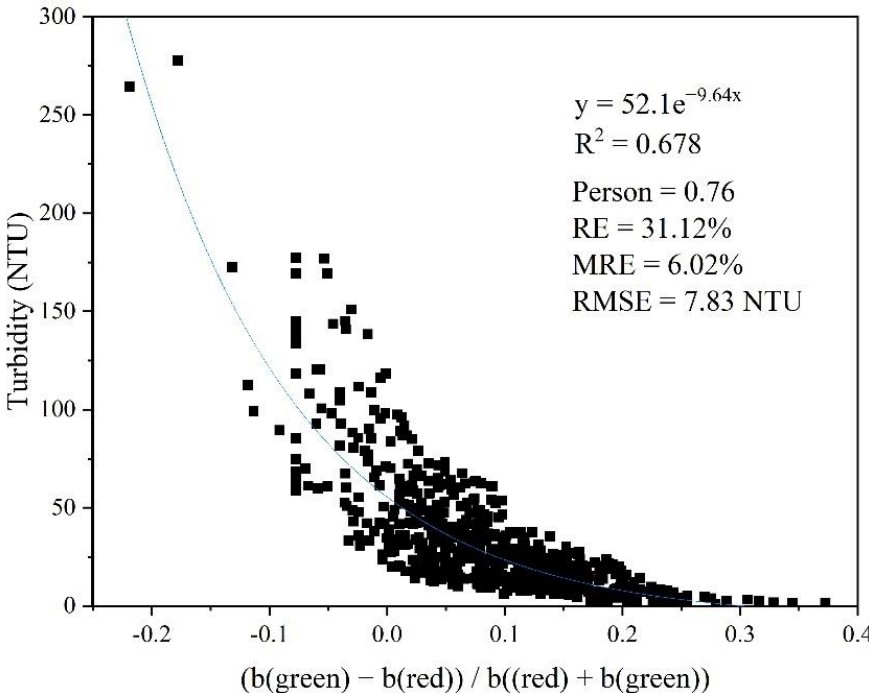

**Figure 6.** Turbidity inversion model.

## 4. Results

### 4.1. Performance of Radiometric Normalization

The turbidity inversion model was applied to the Landsat 7 ETM image both before and after radiation normalization, and then compared with the quasi-synchronous Landsat 8 OLI image. The image used for the test was the Landsat 7 ETM image of the Wuhu section of the Yangtze River, taken on 26 September 2021, while the reference image was the Landsat 8 OLI image of the exact location, taken on 18 September 2021. The results are presented in Figure 7.

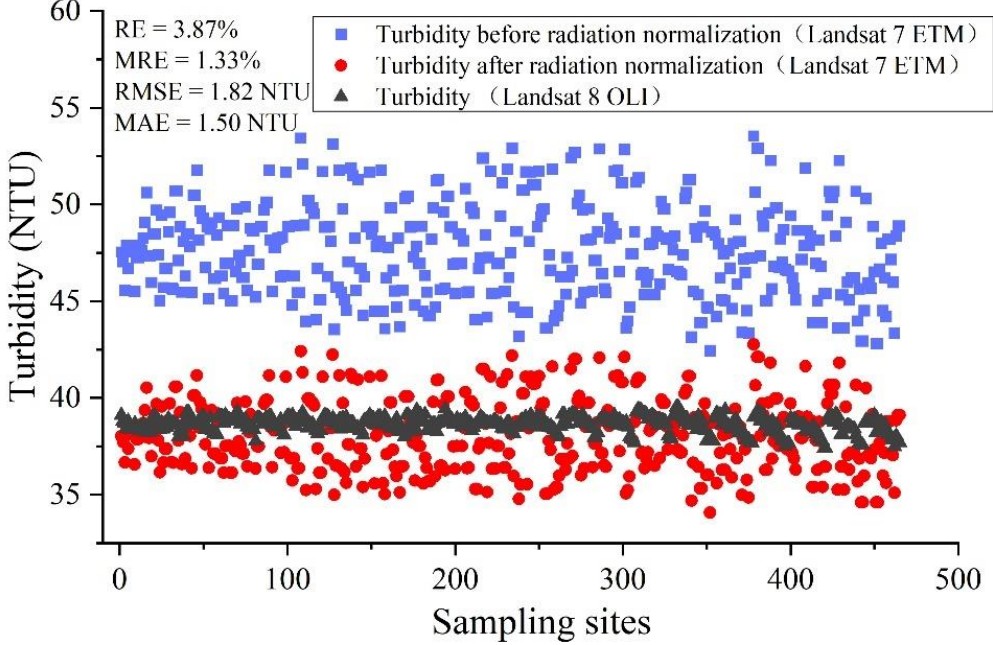

**Figure 7.** Turbidity inversion results and verification before and after radiation normalization.

The overall turbidity inversion results decreased to approximately 40 NTU, which is in line with the results from the reference image. An error analysis of the radiation-normalized turbidity inversion value and the inversion value from the reference image showed that the root means square error for both was only 1.82 NTU, and the average relative error was about 1.33%. In conclusion, the radiation-normalized processing of Landsat ETM/TM image data effectively eliminates differences in sensor response and reduces. It reduces variation, leading to more accurate results in the turbidity inversion.

### 4.2. Turbidity Variations in the Yangtze River Basin at Varied Spatial–Temporal Scales

### 4.2.1. The Entire Watershed

The 36-year average turbidity value in the Yangtze River Basin was calculated to be 40.51 ± 28.62 NTU using the turbidity inversion model. Spatial analysis shows that the turbidity level is generally high in the basin's upper reaches and low in the middle reaches, with higher levels in the main streams and lower levels in the lakes and reservoirs. The frequency distribution of average turbidity in the basin has a positively skewed peak, with 90.8% of water bodies having an average turbidity of less than 70 NTU.

The turbidity inversion results were divided into two periods, the wet period (May–September) and the dry period (January–April and November–December), and the mean turbidity maps were obtained. During the wet period, the average turbidity was 40.41 ± 34.88 NTU, and the range of the water body was larger, with higher turbidity levels in the mainstream and lower levels in the tributaries, lakes, and reservoirs. In the dry period, the average turbidity was 47.54 ± 18.51 NTU, and the range of the water body was smaller, with lower turbidity levels in the mainstream and higher levels in the tributaries and lakes, and reservoirs (as seen in Figure 8).

The monthly average turbidity in the Yangtze River Basin was a seasonal characteristic, with the lowest level in July (33 ± 35.71 NTU) and the highest in February (45.69 ± 27.64 NTU). The average annual turbidity in the basin was calculated on a year-by-year basis and showed a trend of decreasing from 1986–1988 (45.2 NTU) to 2003–2015 (39.3 NTU), with a period of stability from 2016 to 2019 (37.6 NTU), and an increase in 2020–2021 (35.3 to 39.4 NTU). The trend of the annual mean turbidity was generally consistent with the trend of turbidity, with turbidity levels in the dry period being on average 7 NTU higher than in the wet period (as seen in Figure 9).

### 4.2.2. Secondary Watersheds

The mean turbidity values of the secondary basins of the Yangtze River were analyzed (Figure 10a,b). The Yalong River Basin had the highest turbidity level, followed by the Yangtze River main stream Basin. The Wujiang River Basin had the lowest turbidity level. The water bodies were ranked in terms of turbidity as follows: Yalong River Basin > Yangtze River main stream Basin > Minjiang River Basin > Jialing River Basin > Poyang Lake Basin > Dongting Lake Basin > Taihu Lake Basin > Han River Basin > Wujiang River Basin. The difference in turbidity levels between wet and dry periods was minimal in the Yalong and Minjiang rivers. In contrast, the turbidity levels during the wet period were higher in the Yangtze River mainstream Basin and Jialing River Basin and lower in the Hanjiang River Basin, Wujiang River Basin, Dongting Lake Basin, Poyang Lake Basin, and Taihu Lake Basin during the same period.

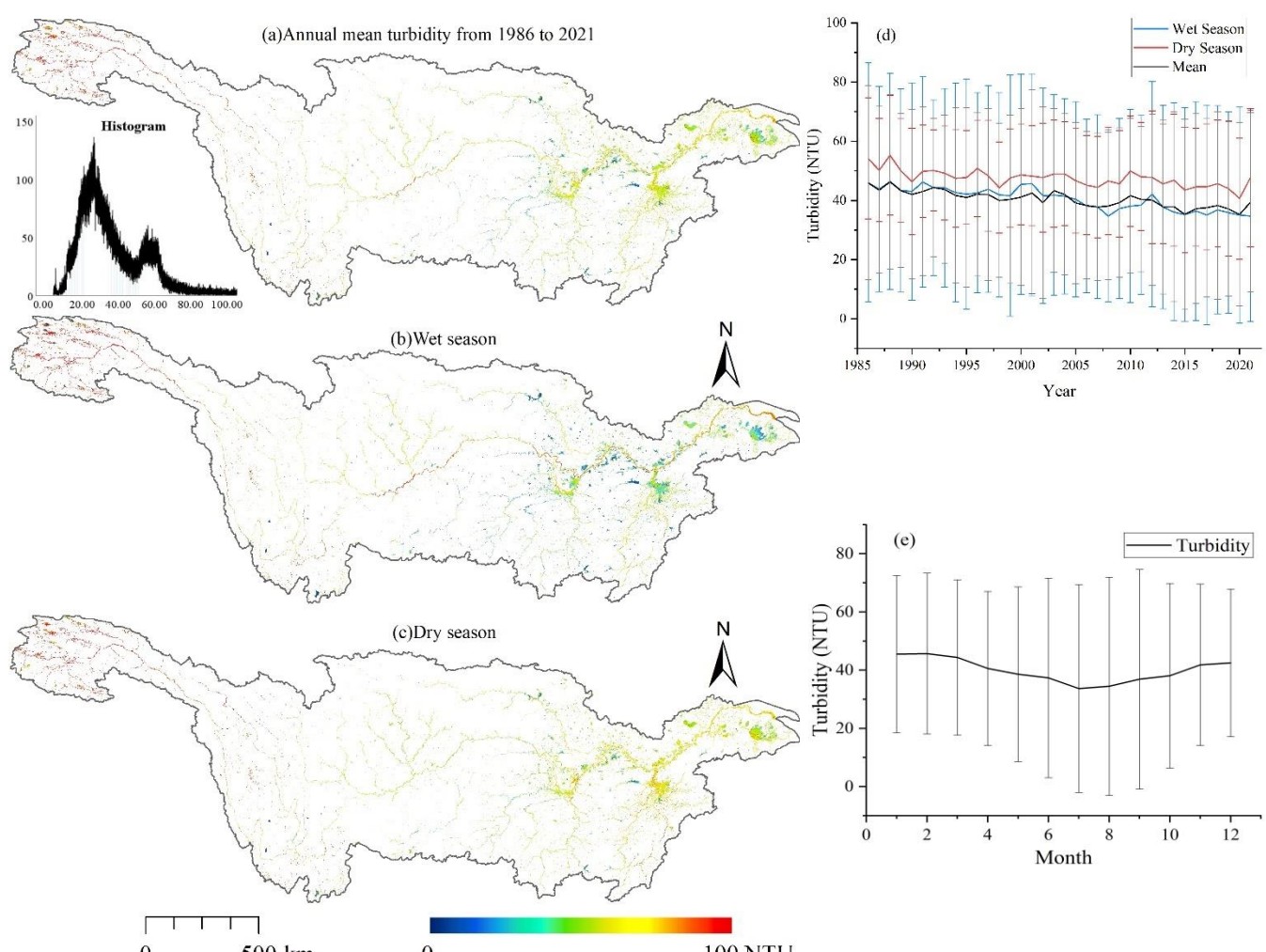

**Figure 8.** Spatial distribution of turbidity in the Yangtze River Basin from 1986 to 2021. (**a**) mean of 36 years and histogram of turbidity; (**b**) wet season; (**c**) dry season; (**d**) annual variations; (**e**) monthly variation.

The statistics of the annual average turbidity and monthly average turbidity of the secondary watersheds (shown in Figure 10c,d) indicate that, except for the Yalong River Basin, which had a constant turbidity level of around 67 NTU in all months except January and December, the overall trend of each watershed has been gradually decreasing over the past 36 years. The Minjiang River Basin, Jialing River Basin, and Yangtze River main stream Basin show a trend of increasing and then decreasing turbidity, with lower levels in spring and winter compared to summer. The Han River Basin, Wujiang River Basin, Dongting Lake Basin, Poyang Lake Basin, and Taihu Lake Basin exhibit a trend of decreasing and then increasing turbidity, with higher levels in spring and winter compared to summer.

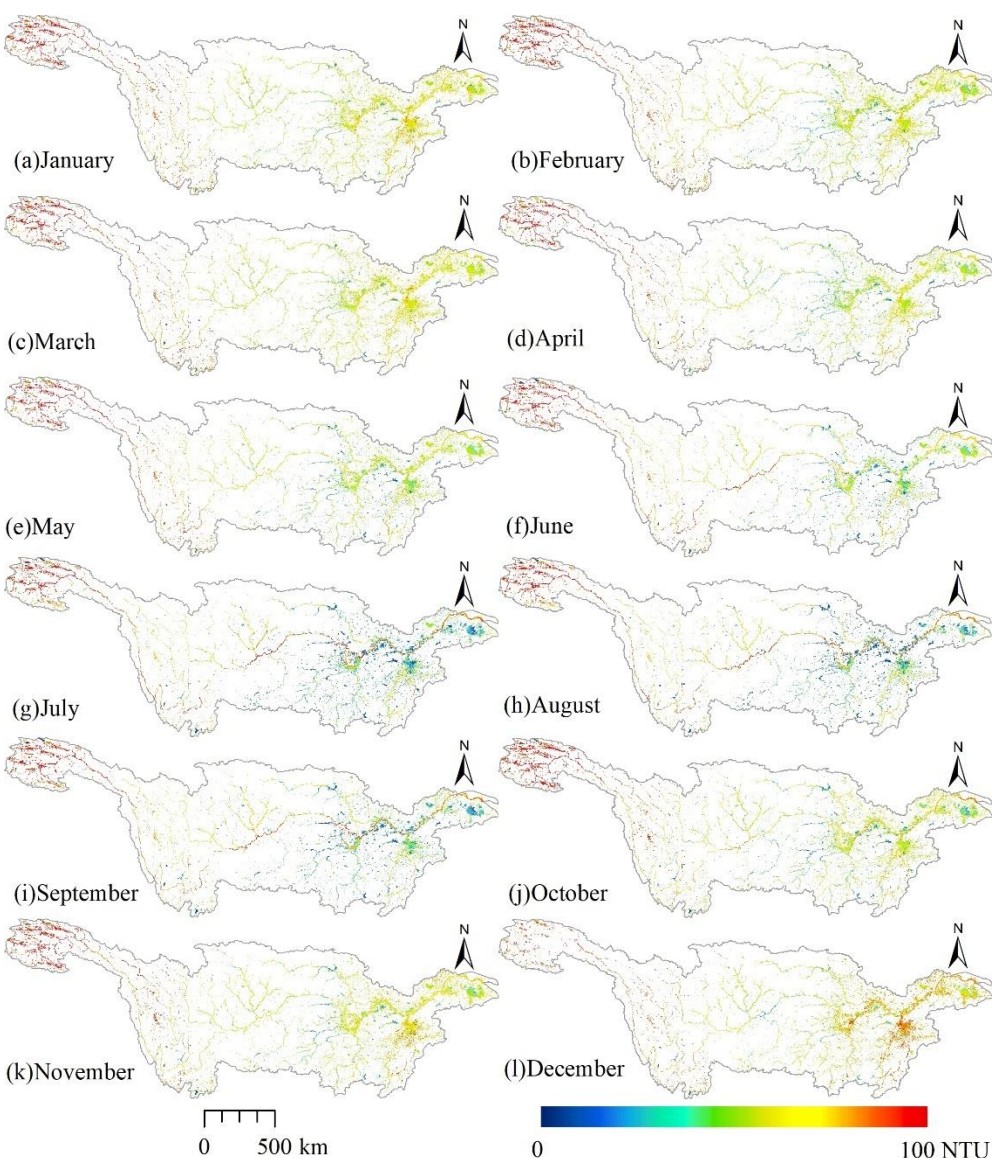

**Figure 9.** Monthly average turbidity distribution in the Yangtze River Basin from 1986 to 2021.

### 4.2.3. Tertiary Watersheds

The 36-year average turbidity values for the tertiary watersheds of the Yangtze River Basin have been calculated (shown in Figure 11). The turbidity levels in the upper reaches of the Yangtze River vary greatly, with the highest being recorded in the Tongtian River (TT) at 81.6 NTU and the lowest in the area above Sinan (SS) at 25.3 NTU, resulting in an average turbidity level of 41.7 NTU and 66% of the sub-basins having 35–45 NTU turbidity levels. In the Yibin to Yichang Main River Basin (YY), the difference between the turbidity levels during the abundant and dry periods is substantial, with an average difference of 33.2 NTU in the wet period compared to the dry period. In contrast, the difference in other sub-basins is minimal, averaging ±6 NTU.

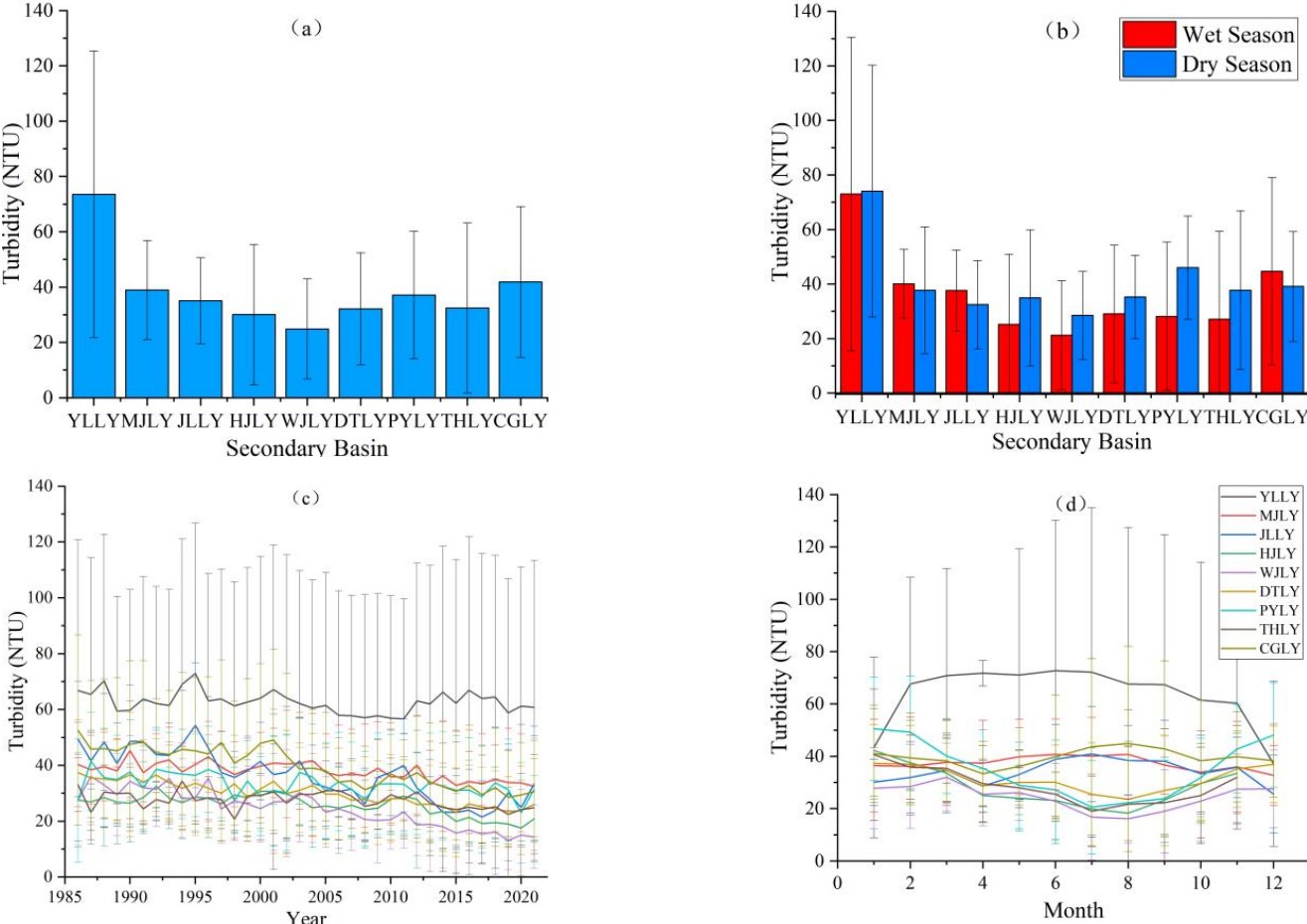

**Figure 10.** Distribution and variation of mean turbidity in the secondary basins in the Yangtze River Basin from 1986 to 2021. (**a**) mean of 36 years; (**b**) mean of wet season and dry season; (**c**) annual variations; (**d**) monthly variations.

In the middle reaches of the Yangtze River, the turbidity levels are relatively low, with the highest level recorded in the Fuxi River (FH) at 40.9 NTU and the lowest in Xiushui (XS) at 19.4 NTU, averaging 30.9 NTU with 78% of the sub-basins having turbidity levels of 26–36 NTU. During the wet period, the turbidity level in the Qingjiang River (QJ), Yichang-Wuhan left bank (YW), and Chenglingji-Hukou right bank (CH) was higher compared to the dry period, while in all other sub-basins, it was higher in the dry period. The highest turbidity level in the lower reaches of the Yangtze River was recorded in the Tongnan and Chongming Island rivers (TC) at 54.9 NTU, and the lowest in the West Lake at 26.8 NTU, averaging 37.6 NTU. The turbidity in the Qingyi and Suyang rivers (QS) and Tongnan and Chongming Island rivers (TC) was higher during the wet period, while in all other sub-basins, it was higher during the dry period.

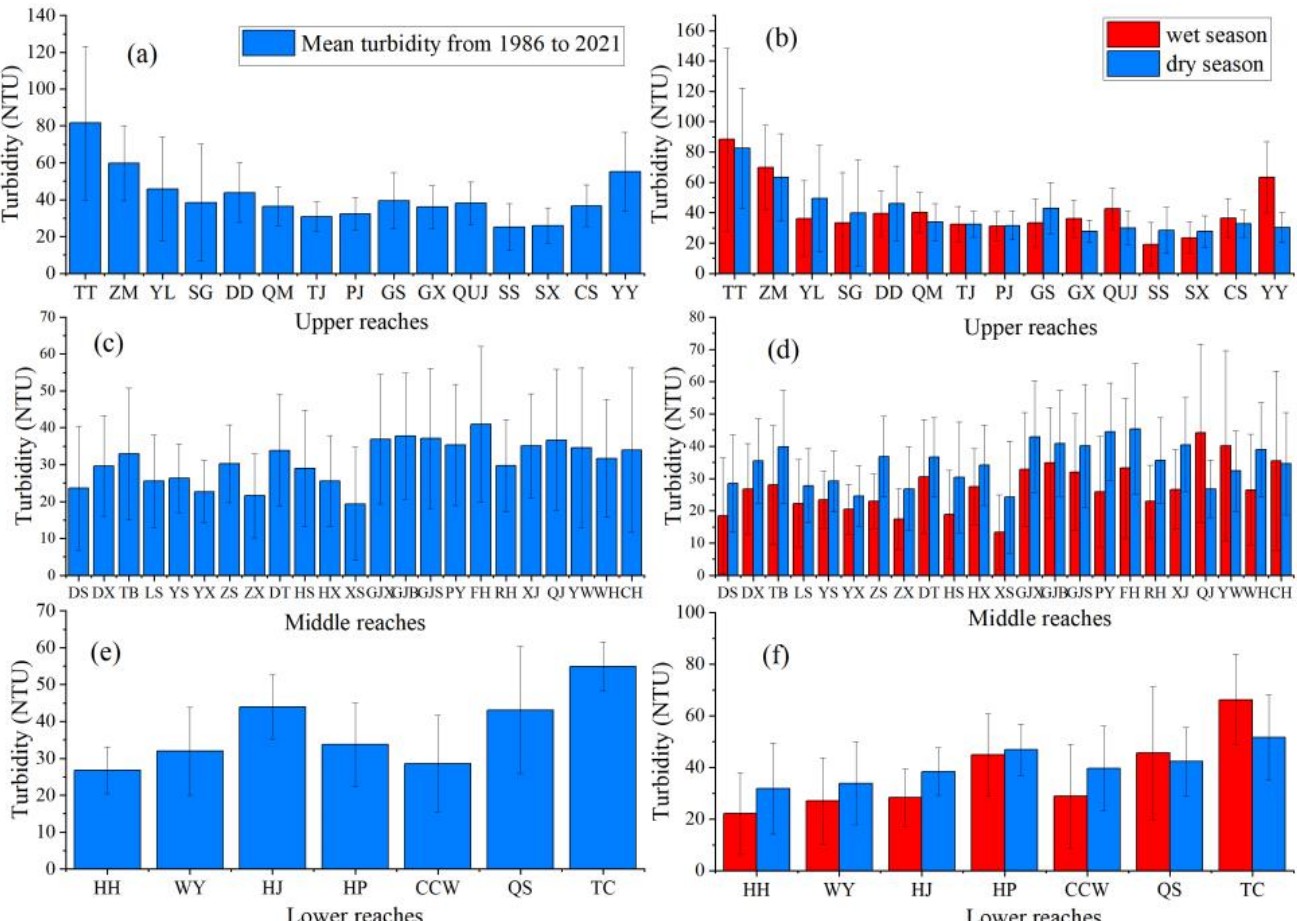

**Figure 11.** Distribution of mean turbidity in the tertiary basins in the Yangtze River Basin from 1986 to 2021. (**a**) mean of 36 years in the upper reaches; (**b**) mean of wet season and dry season in the upper reaches; (**c**) mean of 36 years in the middle reaches; (**d**) mean of wet season and dry season in the middle reaches; (**e**) mean of 36 years in the lower reaches; (**f**) mean of wet season and dry season in the lower reaches.

The annual and monthly average turbidity statistics of the three levels of watersheds (refer to Figure 12) show that the turbidity level of the Tongtian River in the upstream area remained relatively stable, with an upward trend in TT and ZM and a downward trend in all other sub-basins, with the most significant decrease observed in the Yibin-Yichang main stream watershed (YY), which decreased by about 43 NTU. The monthly average turbidity in this sub-basin showed sharp fluctuations throughout the year, with the highest observed from June to August. All sub-basins showed a decreasing trend in the midstream region, with the most significant decrease in the Qingjiang River (QJ) at about 42 NTU. A total of 87% of the sub-basins showed a fluctuating trend, with an increasing trend followed by a decreasing trend during the year, with the Qingjiang River (QJ), Yichang-Wuhan left bank (YW), and Chenglingji-Hukou right bank (CH) showing a fluctuating trend. The Hangjia Lake area (HJ) showed an increasing trend in the lower reaches. In contrast, all other sub-basins showed an oscillating decreasing trend, with the most significant decrease observed in Tongnan and Chongming Island rivers (TC) at 23.4 NTU. The intra-annual trends among the sub-basins were inconsistent, with the most significant changes observed in Tongnan and Chongming Island.

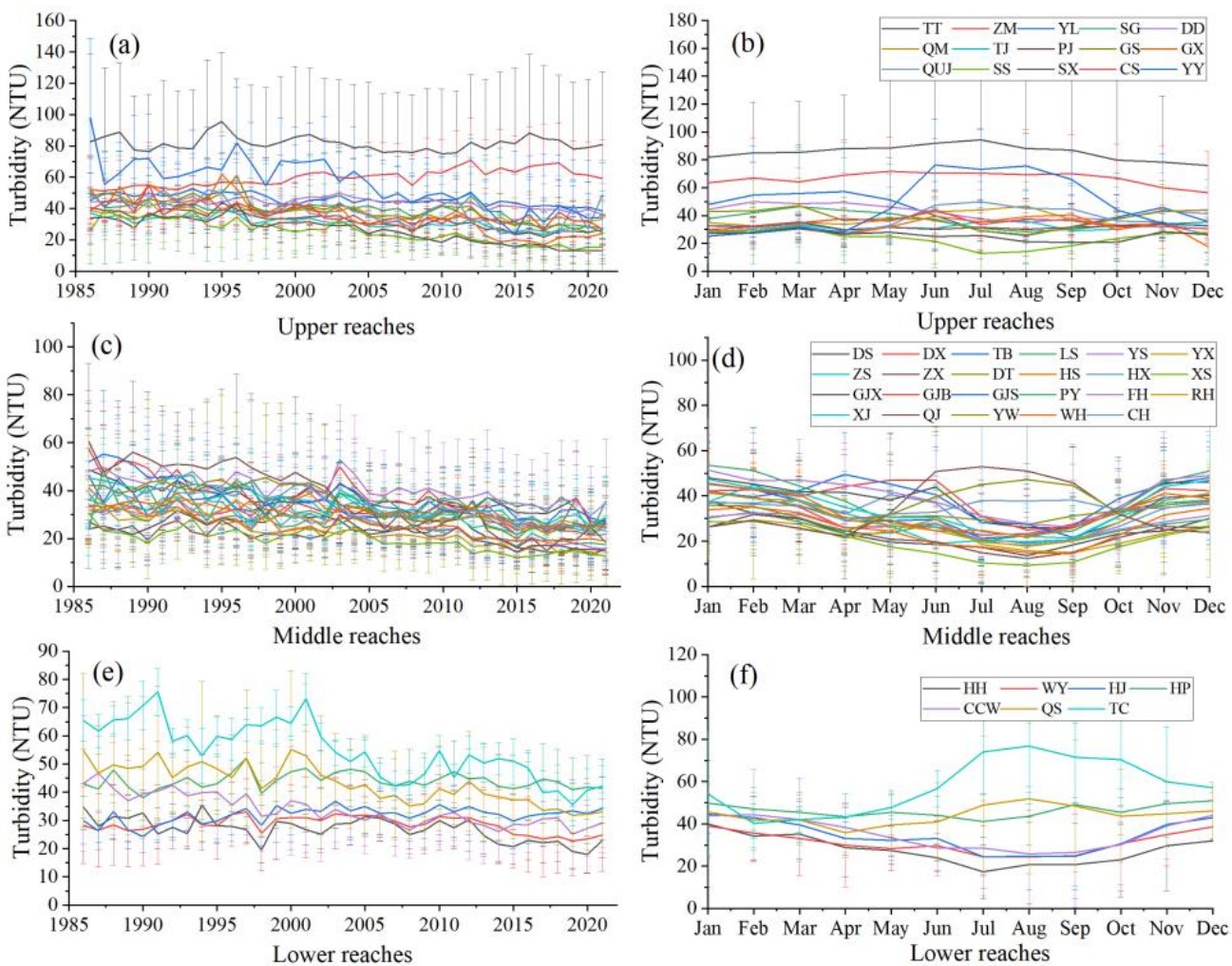

**Figure 12.** Variations of mean turbidity in tertiary basins in the Yangtze River Basin from 1986 to 2021.
(**a**) annual variations in the upper reaches; (**b**) monthly variations in the upper reaches; (**c**) annual
variations in the middle reaches; (**d**) monthly variations in the middle reaches; (**e**) annual variations
in the lower reaches; (**f**) monthly variations in the lower reaches.

The results indicate that the overall trend in turbidity levels in the Yangtze River Basin
is decreasing. The dry period is characterized by higher turbidity levels than the wet period.
The areas with the most significant changes are listed as follows and will be discussed in
the next section: (1) The Tongtian River (TT) and the section from Zhimenda to Shigu (ZM)
in the Yalong River Basin. This area has the highest average turbidity levels in the Yangtze
River Basin, with a significant difference in turbidity levels between lakes and rivers. Lakes
have a decreasing trend of low turbidity levels, while rivers have an increasing trend of
high turbidity levels, and these levels are slightly higher in the abundant water period than
in the dry period. (2) The Yangtze River mainstream Basin (YY, QJ, YW, CH, QS, and TC).
Turbidity levels in this area decrease and then increase from west to east, with the most
significant interannual decreasing trend in the Yangtze River Basin. Additionally, there
is a strong interannual seasonal variation, with higher turbidity levels in the wet period
compared to the dry period. (3) The Hangjia Lake area (HJ). This area has an upward trend
in interannual variation in turbidity levels.

## 5. Discussions

### 5.1. Effects of Precipitation on Turbidity at Short-Long Terms

Precipitation, particularly rainfall, has been identified as a significant factor affecting water turbidity. The effects of precipitation on water turbidity were examined at both annual and monthly scales in the Yangtze River Basin. Several studies have demonstrated a positive correlation between annual precipitation and water turbidity. Increased rainfall can cause soil erosion, which results in higher suspended solids in water. For example, a study conducted in a watershed in Taiwan found that annual turbidity was significantly higher during years with greater precipitation [45]. Similarly, studies in the Middle Reaches of the Yarlung Zangbo River, Southern Tibetan Plateau, showed that river turbidity was affected by the confluence of tributaries and the changes in precipitation and vegetation along the river [46]. However, our results showed a negative correlation between total annual precipitation and turbidity in the Yangtze River Basin over the long term from 1986 to 2021 (Figure 13). In general, the turbidity decreased at the basin and secondary watersheds over 36 years. In contrast, total annual precipitation has increased over the past few decades in most parts of the basin, especially in the upper reaches, such as Yalong River Basin and Minjiang River Basin. This relationship was likely due to the forest recovery policy and action since the 1980s in China, as shown in Figure 4c, where the forest cover increased from 42% to 46% over 36 years. Forest recovery can help prevent soil erosion and reduce the amount of sediment and other particles that enter nearby waterways. The roots of trees and other vegetation can help hold soil in place, and the canopy can help intercept rainfall and slow down surface runoff, allowing more water to infiltrate into the ground. As a result, forest recovery can help improve water quality and reduce water turbidity.

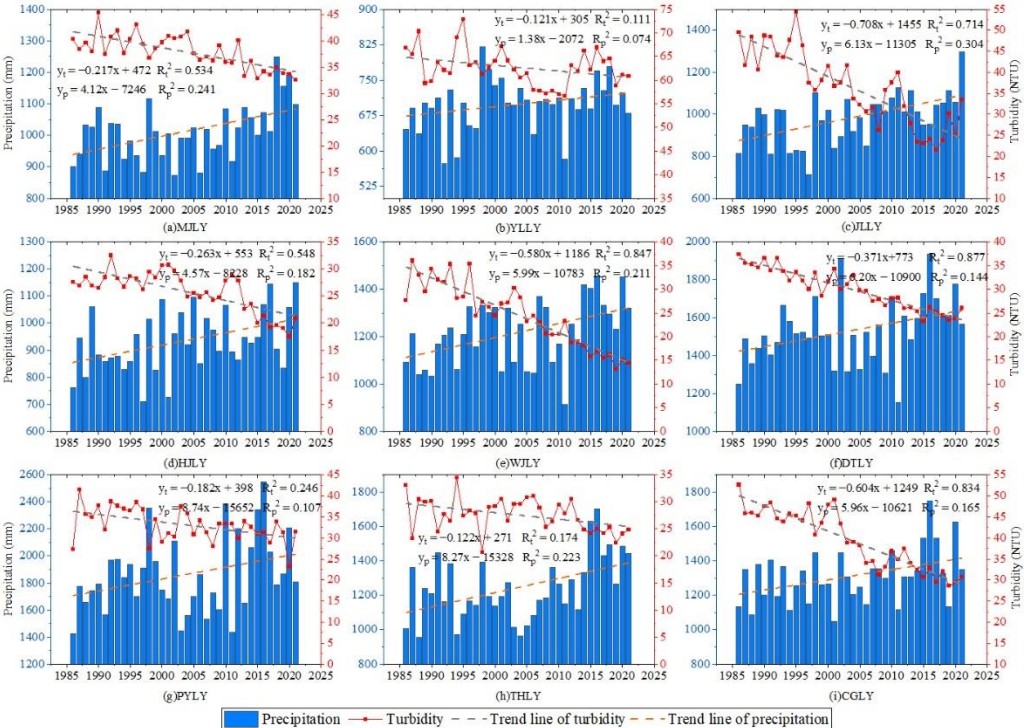

**Figure 13.** Annual precipitation and turbidity variations in the secondary watersheds from 1986 to 2021.

At the monthly scale, our results showed that the relationship between precipitation and turbidity varied depending on the region of interest. Figure 14 presented these relationships over nine secondary watersheds in the Yangtze River Basin. For monthly precipitation, typical seasonal variations were observed for all regions with higher precipitation in summer and autumn, and lower precipitation during winter and spring. However, two different seasonal patterns of water turbidity were found. The first pattern was characterized as "low in summer and high in winter," such as Dongting Lake (DTLY), Poyang Lake (PYLY), and Taihu Lake (THLY), etc., and the second pattern was almost the opposite as "high in summer and low in winter", such as Yalong River Basin (YYLY), Minjiang River Basin (MJLY), and Jialing River Basin (JLLY). The combined effect of both natural factors and human activities may cause this inconsistency. For the regions with the first pattern, we found these watersheds had higher vegetation cover, and larger open waters areas such as lakes rather than rivers (with a mean ratio of lake area to river area of 2.3), which means that during wet seasons with more precipitation, more water will confluence in lakes, and more runoff and sediments will be intercepted, thus reducing lower turbidity in wet seasons. The results were consistent with findings in the middle and lower basins of the Yangtze River, where the largest TSS concentrations occurred in the first and fourth quarters of a year and the lowest values occurred in the third quarter [41]. On the other hand, the Yalong River Basin (YYLY), Minjiang River Basin (MJLY), and Jialing River Basin (JLLY) were found with lower vegetation cover but more rivers (with a mean ratio of lake area to river area of 0.6), thus are more sensitive to precipitation during wet seasons. This pattern may be because during other months, the inputs of sediment and nutrients to water bodies are more strongly influenced by vegetation cover and land use practices rather than precipitation alone.

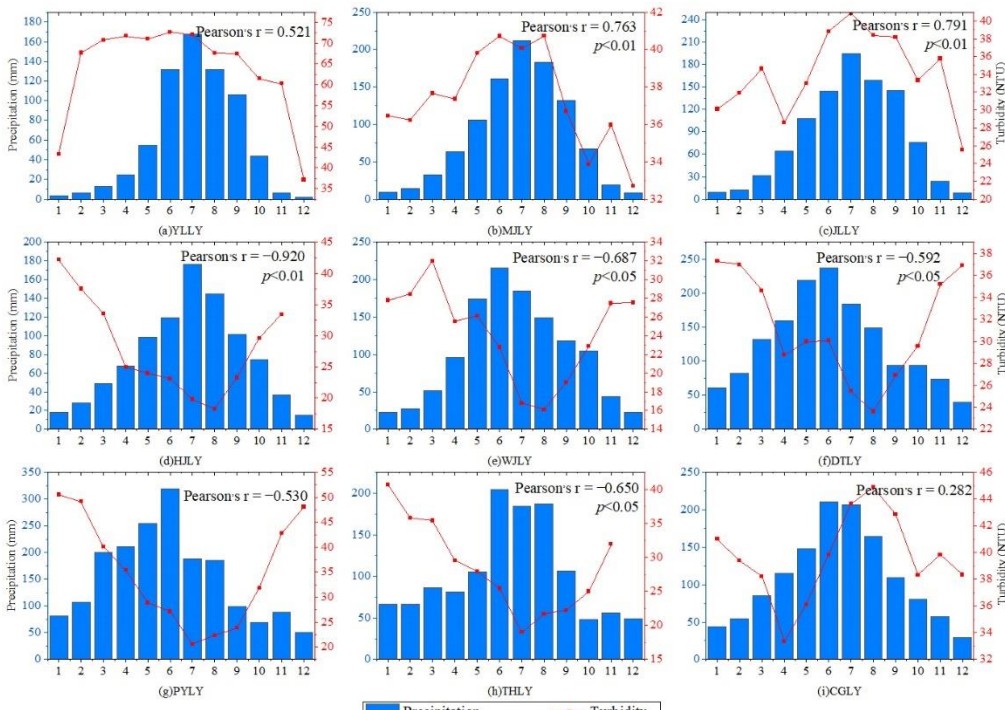

**Figure 14.** Variations of monthly precipitation and turbidity in the secondary watersheds from 1986 to 2021.

### 5.2. Contributions of Varied Drivers on Water Turbidity

The analysis of random forests was conducted to examine the factors affecting the spatial variation of water turbidity in the Yangtze River Basin (Figure 15). The results showed the impact of both natural and human factors on water turbidity was comparable, with natural factors contributing 58% and human activities contributing 42%. Natural vegetation had the most decisive influence, accounting for 43.12% of the total contribution. The impact of these factors differed in different river sections, with natural factors having a greater effect (61%) in the upstream area. In comparison, human activities had a greater impact (54%) in the downstream area.

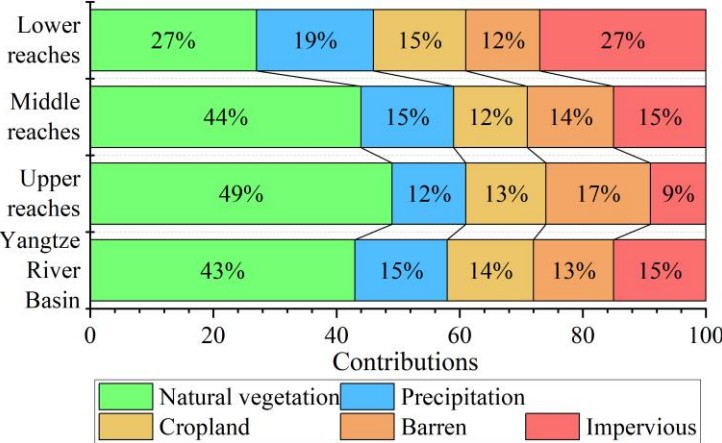

**Figure 15.** Contributions of impacts of explanatory factors on spatial turbidity variations of Yangtze River basin and different reaches using the random forest regression analysis.

A Pearson correlation analysis explored the driving factors of the temporal variation of water turbidity in the Yangtze River Basin (Table 3). The results showed that both natural and human factors influenced the temporal variation of water turbidity, and the variation was negatively correlated with annual and monthly precipitation, positively correlated with cropland and wasteland area, and positively correlated with the increase in impervious area. However, the correlation between turbidity and driving factors varied in regions with abnormal trends.

**Table 3.** Pearson correlation was used to analyze the influence of interpretive factors on the spatial variation of turbidity in the Yangtze River Basin.

| Pearson Correlation Analysis | Yangtze River Basin | TT | ZM | CGLY | HJ |
| --- | --- | --- | --- | --- | --- |
| Precipitation vs. Turbidity | −0.575 ** | −0.260 | 0.125 | −0.417 * | 0.146 |
| Natural vegetation vs. Turbidity | 0.295 | −0.392 * | −0.622 ** | 0.297 | −0.138 |
| Cropland vs. Turbidity | 0.770 ** | −0.443 | −0.033 | 0.901 ** | −0.319 |
| Barren land vs. Turbidity | 0.539 ** | −0.119 | 0.590 ** | 0.876 ** | / |
| Impervious surface vs. Turbidity | −0.801 ** | / | / | −0.909 ** | 0.508 * |

Note(s): * indicates $p < 0.05$, correlation exists; ** indicates $p < 0.01$, correlation is significant.

In TT and ZM, the annual mean turbidity showed an increasing trend and a negative correlation with natural vegetation. The monthly mean turbidity was positively correlated with monthly precipitation. In CGLY, the annual mean turbidity decreased and was significantly associated with changes in cultivated land, wasteland, and impervious area. The monthly average turbidity was positively correlated with monthly precipitation. In HJ, the annual average turbidity showed an increasing trend and was positively correlated with the changes in impervious areas.

### 5.3. Causes of Opposite Turbidity Trends between Mainstream and Lakes

The study shows that both natural factors and human activities play a role in controlling the spatial and temporal variability of water turbidity in the Yangtze River Basin. As an important driver of water turbidity, precipitation contributes to water pollution by transporting surface source pollutants into the river and washing soil particles and pollutants into the river through ground runoff. Accelerated water flow leads to river sediment resuspension, resulting in higher water turbidity levels in summer than in winter, as observed in the Tongtian and Yangtze rivers. However, precipitation also increases the water level of lakes, making them less susceptible to sediment resuspension caused by wind and waves. Dilution of dissolved and suspended substances by rainwater in large lakes such as Poyang Lake, Dongting Lake, and Taihu Lake reduces turbidity, with lower levels observed in summer than in winter. Therefore, lakes in areas with higher precipitation are more likely to have lower turbidity characteristics (Figure 16).

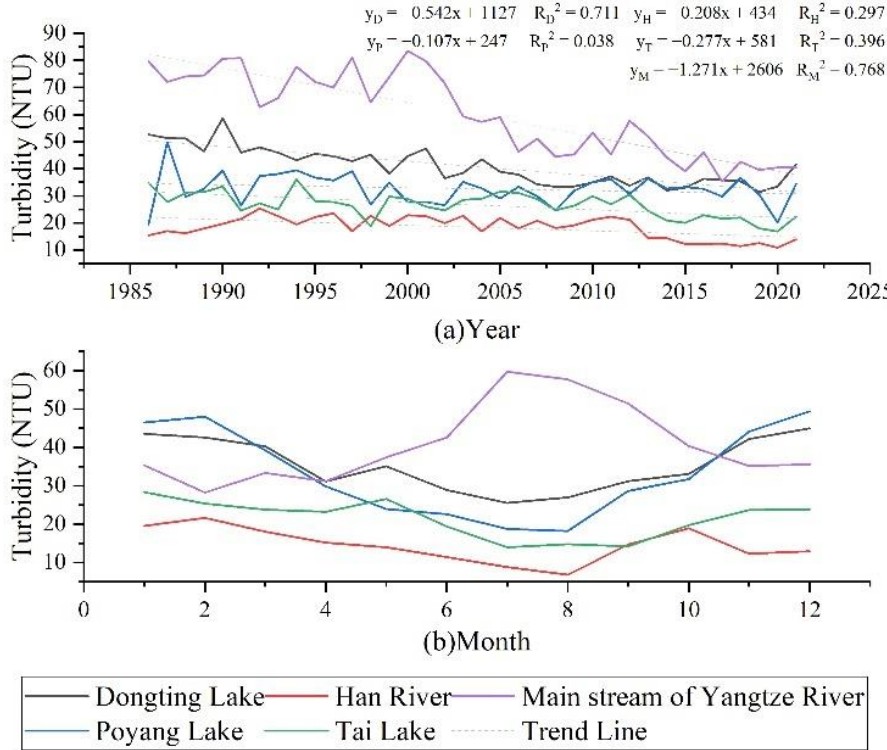

**Figure 16.** Variations of mean turbidity of rivers and lakes in the Yangtze River Basin from 1986 to 2021. (**a**) annual variations; (**b**) monthly variations.

The small proportion of barren land in the Yangtze River Basin accounts for 13% of the spatial variation of water turbidity and significantly affects the time-series variation of water turbidity. Soil erosion in the watershed due to the increase in the area of wasteland leads to a higher concentration of suspended solids in the water column, resulting in higher water turbidity levels. However, higher natural vegetation cover reduces soil erosion and decreases water column turbidity. Human activities, such as agricultural surface pollution, seriously affect the nutrient cycle in the water column, leading to increased eutrophication and higher water turbidity in many lakes. Rapid economic and urbanization development increases pollution, such as industrial and domestic wastewater, resulting in a decline in water quality and higher turbidity levels in the downstream water column with higher urbanization rates. However, under the guidance of the concept of "maintaining a healthy Yangtze River and promoting harmony between people and water" and the Yangtze River Protection Law, provinces and cities in the middle and lower reaches of the river have

started water resources' protection planning, divided units into functional zones, and carried out comprehensive water pollution control.

## 6. Conclusions

Using Landsat images and turbidity inversion models, the spatial and temporal variation of water turbidity in the Yangtze River Basin and its driving factors were analyzed. The overall turbidity trend is decreasing, with higher levels in the upper, a low level in the middle reaches, and a higher level in the river's lower reaches. The Yangtze River Basin shows a decreasing trend, with the most significant decrease in the Yangtze River mainstream. There are local areas with inconsistent changes, such as TT, ZM, and HJ, where interannual changes show an increasing trend. Seasonally, the overall turbidity level in the Yangtze River Basin shows a seasonal variation of "low in summer and high in winter", and the seasonal spatial variation is reflected in rivers and lakes; the turbidity of lakes and reservoirs is "low in summer and high in winter", such as Dongting Lake, Poyang Lake, and Taihu Lake.

Results show that both natural factors and human activities jointly control the variability in water turbidity. Natural factors have a more significant influence on the spatial variation in turbidity, particularly vegetation cover. In contrast, human activities have a more significant impact on the downstream area, with cropland and wasteland positively correlated with turbidity while impervious surfaces are negatively correlated. The negative correlation between water turbidity and annual precipitation over 36 years indicates that the forest recovery policy since the 1980s in China has been effective for water quality management. The study provides a baseline for water environment management in the Yangtze River Basin and provides a reference for remote sensing monitoring of the water environment in inland water bodies.

**Author Contributions:** J.L. proposed the research concept, designed the experiments, conducted data analysis, and composed the manuscript; C.X. designed the framework of this research and participated in manuscript editing. All authors have read and agreed to the published version of the manuscript.

**Funding:** This work was supported by the National Natural Science Foundation of China (No. 41701379, 42071325).

**Data Availability Statement:** The data that support the findings of this study are available on request from the corresponding author, upon reasonable request.

**Acknowledgments:** The authors greatly appreciate the free access of the Landsat data provided by the USGS, and the land cover data provided by the Wuhan University Institute of Remote Sensing.

**Conflicts of Interest:** The authors declare no conflict of interest.

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
