# Peer review of "Drivers of Spatial and Temporal Dynamics in Water Turbidity of China Yangtze River Basin"

_water, doi:10.3390/w15071264_

Round 1
Reviewer 1 Report
The article is titled ‘Drivers of Spatial and Temporal Dynamics in Water Turbidity of China Yangtze River Basin’. The authors used Landsat satellite imagery and in situ observation data to determine the turbidity of the water in the Yangtze River. The spatial and temporal dynamics of turbidity and the factors driving it were investigated. The article is interesting, but needs improvement. My comments are as follows:
- the abstract needs to be shortened
- The authors use abbreviations, e.g. NTU, they do not explain the name on first use, e.g. line 14 and 15
- Fig. 1 requires improvement, the Yangtze River is not marked, catchment markings are misleading, no reference to a larger region, e.g. additionally marking China or Asia on the contour map, legend needs improvement, e.g. monitoring sites or monitoring stations? The map with precipitation needs improvement, only 4 ranges of values were used, in the text the authors write about precipitation in the amount of 1300 mm, which is not visible on the map, there is no such range (fig. 3a should be corrected, the scale on the map should be given in km, not miles), in Fig. 3b there is no given data source
- No information on when the dry season starts and when the rainy season begins. There is only information when there is maximum rainfall (line 189-190)
- research methodology is not clearly described, present it in the form of a step-by-step block diagram
- mark what is new in the article
Technical Notes:
- fig 7 is unclear,
- line 14 is R2 and should be R2
- the list of references should be adapted to the requirements of the journal
- no description of the author's contribution (line 529), which is mandatory, funding (line 530)…
- the article requires extensive language editing, e.g. number of sites or images?
Reviewer 2 Report
The article is written in an intelligible manner and can be useful for readers.
A few observations are found below.
In the introduction section: "A multi-band combination model is constructed to invert water turbidity, and the model’s accuracy is analyzed. The study also examines the spatial and temporal distribution characteristics of water quality parameters based on the inversion resul." You should rephrase these lines in order to detail what the "invert" and "inversion" are or to remove such details which need better despription in order to avoid misinterpretation at this point of the manuscript.
Figure 1 - the abbreviation of each color in the legend should be detailed into the figure caption or make a reference to table 1.
The coloured dots in the maps of figures 7 and 8 could be bigger - these maps might need a reworking as there is a lot of useless white space.
Figures 10 + 11 - "Distribution of mean turbidity in the secondary Yangtze River Basin" - secondary is an unusual term in this position, better replace with something like "of the secondary/tertiary basins in the Yangtze River Basin".
Round 2
Reviewer 1 Report
The authors made corrections to the article in accordance with the reviewer's comments. The article is ready for publication.